# Should We Not Further Study the Impact of Microbial Activity on Snow and Polar Atmospheric Chemistry?

**DOI:** 10.3390/microorganisms7080260

**Published:** 2019-08-14

**Authors:** Florent Domine

**Affiliations:** 1Takuvik Joint International Laboratory, Université Laval (Canada) and CNRS-INSU (France), Québec, QC G1V 0A6, Canada; florent.domine@gmail.com; 2Centre d’Études Nordiques (CEN), Department of Chemistry and Department of Geography, Université Laval, Québec, QC G1V 0A6, Canada

**Keywords:** snow, chemistry, microbes, atmosphere, climate

## Abstract

Since 1999, atmospheric and snow chemists have shown that snow is a very active photochemical reactor that releases reactive gaseous species to the atmosphere including nitrogen oxides, hydrocarbons, aldehydes, halocarbons, carboxylic acids and mercury. Snow photochemistry therefore affects the formation of ozone, a potent greenhouse gas, and of aerosols, which affect the radiative budget of the planet and, therefore, its climate. In parallel, microbiologists have investigated microbes in snow, identified and quantified species, and sometimes discussed their nutrient supplies and metabolism, implicitly acknowledging that microbes could modify snow chemical composition. However, it is only in the past 10 years that a small number of studies have revealed that microbial activity in cold snow (< 0 °C, in the absence of significant amounts of liquid water) could lead to the release of nitrogen oxides, halocarbons, and mercury into the atmosphere. I argue here that microbes may have a significant effect on snow and atmospheric composition, especially during the polar night when photochemistry is shut off. Collaborative studies between microbiologists and snow and atmospheric chemists are needed to investigate this little-explored field.

## 1. Introduction

Microbes are ubiquitous on Earth and have colonized the most hostile and improbable environments, including snow [1], glacial ice [2] and permafrost [3]. Microbiologists have studied their adaptations and how various environments imposed them onto these small living creatures [4]. Interactions between various microbes and between microbes and other organisms have also been extensively studied [5]. However, how microbes impact the chemistry and the physics of their environment have received comparatively little attention. The case of snow chemistry is particularly interesting because changes in snow chemical composition, whether caused by photochemistry, dark chemistry, or microbes, also have the potential to affect atmospheric composition and hence climate [6,7], as well as the carbon budget of the cryosphere [8,9]. Regarding snow studies, microbiologists have expectedly focused on their field and have characterized microbial assemblages and quantified populations [10,11]. They have considered the availability of nutrients but have paid little attention to the impact of microbial metabolism on snow chemical composition and reactivity. In parallel, snow chemists have likewise focused on their own field and interpreted the chemical composition and reactivity of the cryosphere in terms of chemical reactions [12] while often ignoring the possible impact of microorganisms. Arguably, too few interactions have taken place between these two scientific communities. The purpose of this short note, which is not an exhaustive review, is to illustrate with selected examples that snow and atmospheric chemistry may be significantly dependent on microbial activity. My purpose is to stress the need for greater interactions between snow chemists and microbiologist to understand these issues, which also have implications for the climate. Indeed, in the current climate change context, the cryosphere and in particular the snow is one of the most affected media [13]. I therefore argue that understanding how climate change will impact microorganisms and how these may in return feedback on the climate through their effect on snow and atmospheric chemistry requires highly interdisciplinary collaborations.

## 2. Biologists Detect and Identify Microbes in Snow

Microbiologists have been quantifying and identifying microbes in cold snow for decades. Carpenter et al. [1] quantified bacteria in the snow at South Pole by filtering and dying melted snow. Using DNA extraction and 16S rRNA sequencing, they determined that most of them were related to the *Deinococcus* group. Interestingly they performed incubations at sub-freezing temperatures, between −12 and −17 °C using tritiated thymidine and leucine, and they observed “low rates of bacterial DNA and protein synthesis.” However, they did not investigate how this activity impacted snow or atmospheric chemistry. The detection of metabolic activity in cold snow, rather than in melting snow as done in most snow-related studies, is important because most snowpacks on Earth are cold, i.e., do not contain appreciable amounts of liquid water.

Segawa et al. [11] investigated bacteria in temperate snow in Japan using methods essentially similar to those of Carpenter et al. [1] and stressed the fact that their numbers considerably increased during snow melt, which lasted months in this region which is affected by very heavy snowfalls. As above, the impact of bacteria on snow composition was not studied.

Amato et al. [14] studied bacteria in Svalbard snow and did mention that bacteria “have the capacity to degrade organic compounds found in Arctic snow (propionate, acetate and formate),” implicitly stating that this would alter snow composition. They did, however, mention that this could be important when snow melts. They were more interested in bacterial growth during melting than in the impact of this growth on environmental chemistry.

Mercury in snow has also drawn the attention of microbiologists. This element has a long atmospheric residence time and is transported to polar regions. It episodically reaches peak concentrations in polar snow because of atmospheric mercury depletion events (AMDEs) where atmospheric Hg^0^ is massively oxidized to Hg^II^ [15] by halogen compounds such as Br atoms. This Hg^II^ attaches to aerosols and is deposited to snow. How microbes react to the extra inputs of this contaminant to snow has been investigated by Møller et al. [16]. They performed one of the few studies where the impact of bacterial metabolism on snow chemical reactivity was considered. They focused on mercury-resistant bacteria which included Alpha-, Beta- and Gammaproteobacteria, Bacteriodetes, Firmicutes and Actinobacteria. The resistance mechanism is by the reduction of Hg^II^ to Hg^0^, which is volatile and is released to the atmosphere. This is important because the air–snow exchange of mercury is a very active chemical process in the Arctic, and it affects both atmospheric composition and the biosphere [17]. By making several assumptions, they estimated from their bacterial counts that, on average, bacterial activity was responsible for 2% of snowpack emissions of Hg^0^.

Ariya et al. [18] showed interest in the impact of bacteria on snow composition in organics. They analyzed volatile organics in urban snow but found no correlation with bacterial counts. Pursuing their investigations, they incubated melted snow doped with ^13^C-labelled malonic acid at 4 °C and used NMR to detect the formation of organic metabolites in snow. However, they did not test bacterial activity in cold snow.

Larose et al. [19] studied nitrogen metabolism in Svalbard snow in spring. They monitored snow chemical composition (ammonium, nitrate, and nitrite), as well as bacterial types and genomes. They found genes for all steps of the N cycle except for nitrification (NH_4_^+^ to NO_2_^−^ to NO_3_^−^), implying that bacterial metabolism probably affected snow composition. They showed the co-variation of selected bacterial types with relevant chemical species (e.g., NH_4_^+^ and *Nitrobacter*; NO_3_^−^ and *Bacillus*) but did not quantitatively discuss how bacterial metabolism modified the composition of the snow. They did not mention the snow temperature but reported that the snowpack was ripe (i.e. isothermal at 0 °C so that liquid water was ubiquitous) around the middle of their two-months campaign, suggesting that their results were affected by liquid water.

It is worth mentioning that the paucity of studies on the impact of bacterial metabolism on snow chemical composition somewhat contrasts with similar studies on glacial ice. Ice cores are unique environmental archives which have therefore been the subject of intensive chemical analyses. At the turn of the century, Sowers [20] analyzed N_2_O variations in the Vostok ice core during the penultimate deglaciation, around 130,000 years before present. They found some N_2_O concentrations suspiciously higher than present. These were associated with high dust levels and cell counts (determined from the epifluorescence microscopic examination of melted ice core samples) which led him to propose that “One possible explanation for the anomalous N_2_O data is in situ N_2_O production by nitrifying bacteria.” He performed N_2_O isotopic measurements which were consistent with this hypothesis. This seminal work demonstrated the potential impact of microbial metabolism on the chemistry of ice caps even at very cold temperatures. Subsequently, Price and Sowers [21] argued that microbes could remain metabolically active in ice at temperatures lower than −40 °C, although the actual detection of metabolic activity in such cold conditions would be problematic, since they predicted just 10 carbon turnovers in one billion years. In following studies, artefacts in concentrations of CO, N_2_O, and CH_4_ in ice cores from Greenland or Antarctica were attributed to microbial activity, see e.g. [2,22,23] and references therein. However, though observing chemical effects of microbes over timescales of tens to hundreds of thousands years in ice caps is important for ice core interpretation, it has little impact for current environmental chemistry. On the contrary, observing the microbial production of chemical species in cold snowpacks impacts current environmental chemistry. Snow chemistry has obviously been studied by chemists, who have detected chemical reactions but have, in general, showed little interest for biological activity.

## 3. Chemists Detect Photochemical Reactions in Snow

Snow chemical composition has interested environmental chemists since at least the 1960s, as it allowed for the study of pollutant transport [24] and geochemical cycles [25]. Yet, for decades, snow was essentially considered an inert reservoir of trace chemical species until snowmelt. Things changed in 1999 when two studies revolutionized the way we viewed snow chemistry. Honrath et al. [26] discovered in Greenland that the nitrate ion NO_3_^−^ contained in snow was photolyzed to produce NO_x_ (= NO + NO_2_), which were released to the atmosphere. Since NO_x_ are key species in atmospheric chemistry [27], chemical reactions in snow were found to considerably affect tropospheric chemistry. For example, NO_x_ production can lead to the formation of ozone, a pollutant and strong greenhouse gas. The simultaneous work of Sumner and Shepson [28] led to a similar conclusion. They observed in the Canadian high Arctic that formaldehyde, HCHO, was produced in snowpacks by the photolysis of organic precursors and released to the atmosphere. Since HCHO is a source of OH, a key atmospheric oxidant, this finding also had an immense impact. Subsequently, other studies in the Arctic [29] and Antarctic [30] confirmed the photochemical activity of snow. These studies observed the photochemical emission of numerous species such as hydrocarbons, methyl halides and of course, mercury [31]. Snow then started to be viewed as a photochemical reactor capable of critically affecting the chemistry of the lower atmosphere [7]. This is illustrated in Figure 1. By releasing compounds such as organics that could subsequently be oxidized to less volatiles species capable of forming aerosols, snow photochemistry also has an impact on the climate. Indeed, aerosols both scatter light and act as cloud condensation nuclei, thus changing cloud droplet size and hence albedo [32]. Therefore, snow photochemistry studies have widespread applications. However, even though models for snow chemistry have been designed [33], all of them by environmental chemists, none of them include microbial activity.

## 4. Chemists Detect Impact of Microbial Metabolism in Snow

In fact, most snow chemists have continued, rather unsurprisingly, to describe snow chemistry in terms of chemical reactions only. More than a decade after the key findings of Honrath et al. [26] and Sumner and Shepson [28], a review of chemical processes in snow [37] barely mentioned the potential role of microbes in snow chemistry, and the reviewers of that paper, most likely chemists, were not the least bothered by this deficiency. This is a bit troubling, given that an earlier study [6] had observed emissions of NO_x_ from the snowpack in Svalbard during the polar night. Since nitrate photolysis had to be ruled out to explain the source of NO_x_, the authors had no choice but to invoke the role of nitrifying and denitrifying bacteria in those emissions (Figure 1). It is noteworthy that the initial Amoroso team [6] was comprised of only snow and atmospheric chemists and physicists, but they recruited biologists after their campaign to refine their interpretation, demonstrating the need for collaboration. The snow studied by Amoroso et al. [6] was cold, with temperatures always being <-10°C. This is interesting, since, with the exception of the work by Carpenter et al. [1], investigations by biologists of metabolic activity in snow were mostly at or near 0 °C where liquid water is present.

That liquid water was not needed for microbes to have a rapid impact on snow and tropospheric chemistry indicates that polar regions may in general be concerned by microbiological processes in snow. The NO_x_ winter fluxes measured by Amoroso et al. [6] were of the same order of magnitude as those detected by Honrath et al. [26], showing that microbial and chemical activity in snow deserved about equal consideration for understanding the dynamics of snow composition and consequently that of the lower polar troposphere.

Confirming microbiological activity in snow, Antony et al. [9,38] performed a detailed chemical analysis of organic compounds in coastal and near-coastal Antarctic snow using ultra-high resolution mass spectrometry (FTICR-MS). They concluded that organic compounds had been extensively transformed by microbial activity, and they detected both degradation and synthesis reactions.

Redeker et al. [39] investigated the exchanges of methyl iodide between snow and atmosphere in Svalbard and at Signy Island, Antarctica. To determine the possible role of microbes, they worked on snow irradiated with UV-C (for sterilization) and on untreated snow. They noted that irradiated snow was a sink for CH_3_I while non-irradiated snow was a very slight source of this compound, from which they deduced that microbes in snow produced CH_3_I. It is not certain, however, that the snow studied was not melting, since the air temperature was close to 0 °C, and accurately measuring surface snow temperature in the presence of solar radiation is a delicate process. Incidentally, Redeker et al. [39] ignored the previous work of Amoroso et al. [6] and of Antony et al. [38], and they wrongfully claimed that they “present the first evidence of environmental alteration due to in situ microbial metabolism of trace gases … in polar snow.” I report this omission here to illustrate how many snow chemists are little aware of microbial activity in snowpacks.

## 5. The Need for Collaborations

The work of Amoroso et al. [6] in the high Arctic showed that during the polar night microbial activity could be a source of NO_x_ to the polar troposphere. This was quite a novel finding because at that time it was considered that microbial emissions of NO_x_ from non-photochemical sources had to come from soils. Since the ground was frozen, it was implicitly understood that there was no significant local source of NO_x_ to the polar troposphere. Likewise, the possibility exists that microbes in snow could release hydrocarbons [9], halocarbons [39], aldehydes, mercury [16], etc., thus modifying both snow and atmospheric composition (Figure 1). Campaigns to study snow and atmospheric chemistry in polar regions usually take place in spring or summer and their focus is on photochemistry. Photochemical processes are probably predominant over microbial ones in most cases under sunlit conditions, as suggested by Møller et al. [16] for mercury. However, during the long polar night, tens of millions of km^2^ are snow-covered, both on land and sea ice, and could act as a source of reactive molecules to the atmosphere. Both chemical and microbiological studies are probably required to detect these processes and elucidate their mechanisms. Isotopic tools are very efficient to detect microbial activity: In the end, it is isotopic data that demonstrated the microbial production of N_2_O in ice cores [20] and of NO_x_ in polar snow [6]. Identifying microbes and detecting potentially active genes likely to result in significant modifications of snow and atmospheric chemistry appears as a worthwhile goal. Cold incubations, as done by Carpenter et al. [1], may reveal unsuspected processes, especially if they are associated with chemical studies investigating changes in snow and air composition.

The study of N metabolism in snow illustrates the potential benefit of collaboration. Amoroso et al. [6] and Larose et al. [19] studied the chemical and microbial aspects of N (bio)chemistry in Svalbard snow. Both studies took place independently, two years apart, at the same site. Amoroso et al. [6] worked during the polar night, in the total absence of radiation. Using NO_2_^−^ and NO_3_^−^ isotopes (δ^15^N and Δ^17^O), they unambiguously demonstrated that the concentrations of these ions in snow had a strong contribution from nitrifying microbes and that subsequent microbial denitrification or nitrification explained the observed emissions of NO to the atmosphere. For chemists, this was rather revolutionary, since they had always explained the concentrations of NO_3_^−^ in snow exclusively in terms of chemical processes [33,40]. Of course, Amoroso et al. [6] did not use metagenomics or metatranscriptomics to identify the microbes and active genes involved, which would have made for a uniquely holistic approach. Larose et al. [19] worked in spring (24 h sunlight) and studied the snow metagenome. They found no reads associated with nitrification, in contradiction with the results of Amoroso et al. [6]. They did find reads associated with denitrification, which then probably resulted in NO and N_2_O emissions to the atmosphere, but these species were not measured during this essentially microbiological study. Even though Larose et al. [19] analyzed nitrogen ions in snow, their chemical studies were not sufficient to reach critical conclusions regarding snow and atmospheric chemistry. For example, NO_3_^−^ isotopes would have been useful to confirm the absence of nitrification. Furthermore, the atmospheric measurements of NO and NO_2_, coupled to shading experiments to remove the effect of NO_3_^−^ photochemistry, may have quantified bacterial N oxides emissions. Joint studies may also have explained why nitrification was observed by Amoroso et al. [6] in winter and not by Larose et al. [19] in spring. Perhaps the oxidative stress induced by photochemistry played a role [10] in suppressing nitrification, but this is only speculation.

N metabolism is just one example. The metabolism of carbon, mercury and many other elements deserves our attention. Antony et al. appear to have performed pioneering studies of carbon metabolism in snow. They studied the detailed composition of organics in snow [9,38] and in a separate study identified microbes and some of their enzymatic activity [41]. However, the detailed understanding of the coupling between microbial activity and changes in snow chemical composition still has to be reached. Eventually, multi-omics methods coupled to state of the art chemical analysis such as FTICR-MS may reveal how microbes modify the chemical composition of snow and perhaps how this impacts the composition of the atmosphere and climate. Such complex methods have been successfully applied to other fields such as marine chemistry [42,43,44] with significant success, and they would clearly benefit snow biochemical studies.

With climate warming, the presence of liquid water in polar snowpacks may happen more often in winter [45]. This could lead to very large microbial population growth [11,14] and therefore to a large amplification of microbial activity in polar snowpacks. This, in turn, could induce significant modifications of both snow and atmospheric composition and, therefore, climate. The composition of seasonal snow also may have a large impact on terrestrial and marine ecosystems. For example, Hood et al. [46] investigated organic matter (OM) input from melting snow and glaciers into the gulf of Alaska. They found that this OM was preferentially metabolized by aquatic heterotrophs. They concluded that “glacial runoff is a quantitatively important source of labile reduced carbon to marine ecosystems.” Therefore, the way microbes affect OM composition in snow may impact the oceanic and arguably the terrestrial food web. Since this may be modified by climate change, investigating the interactions between snow microbiology and snow and atmospheric chemistry may reveal important snow–biosphere–climate feedbacks mediated by snow microbes.

## Figures and Tables

**Figure 1 microorganisms-07-00260-f001:**
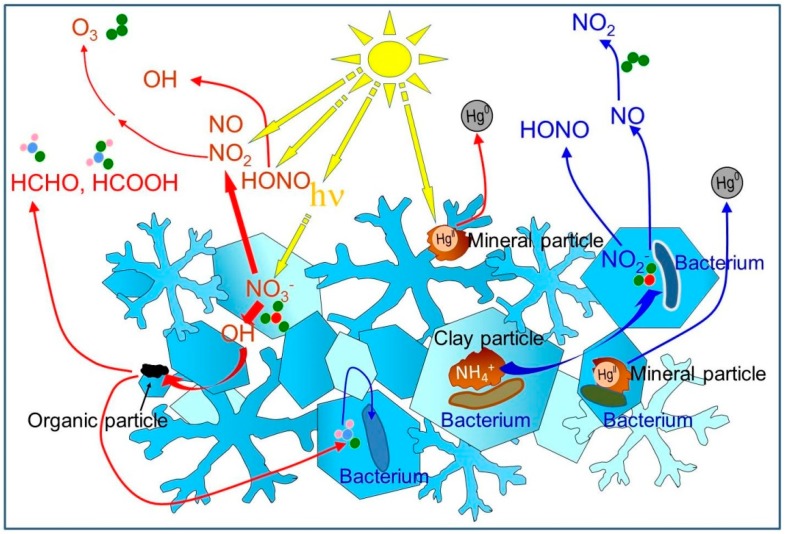
Selected observed and hypothetical modifications in snow and atmospheric composition due to photochemistry (red arrows) and microbial metabolism (blue arrows). Photochemical reactions photolyze the nitrate ion, producing NO, NO_2_, and HONO. Secondary chemistry produces OH, which can oxidize organics present on organic particles [7]. This chemistry releases NO, NO_2_, HONO, HCHO, and other organics to the atmosphere [18,28]. HCHO can form a solid solution with ice [34], thus becoming incorporated within snow crystals. Photochemistry reduces Hg^II^ to Hg^0^, which is released to the atmosphere [17]. Gammaproteobacteria also reduce Hg^II^ to Hg^0^ [16]. The nitrifying bacterium *Nitrosomonas* produces NO_2_^−^ from NH_4_^+^ contained in clay mineral particles deposited onto snow by wind. Under suitable pH conditions (snow is often acidic), NO_2_^−^ can be released to the atmosphere as HONO. The denitrifying bacterium *Pseudomonas stutzeri* produces NO from NO_2_^−^ [6]. *Nitrosomonas* can also denitrify and produce NO. NO can be oxidized to NO_2_ by atmospheric ozone. As suspected from ice core analyses [35], bacteria embedded within the ice lattice may also metabolize molecules that diffuse in ice and I speculatively represent here the consumption of HCHO by *Methylobacterium* [36].

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
