# Peer review of "Should We Not Further Study the Impact of Microbial Activity on Snow and Polar Atmospheric Chemistry?"

_microorganisms, 2019, doi:10.3390/microorganisms7080260_

Round 1
Reviewer 1 Report
The present postion paper emphasizes the need to strenghten collaborations between microbiologists and chemists to elucidate the effect of sympagic microbial communities on the atmospheric chemistry over snow and ice covered areas. It is very welcome contribution in times when the polar and snow-covered areas around the globe undergo profound changes. Hopefully, it is not too late for the results of these collaborations to bear fruit.
The paper in its present form is very good and may be published as is. However, here are three suggestions, which might in my opinion improve the context and understanding for the reader. First, I would invite the author to consider adding to the manuscript the phenomenon of Atmospheric Mercury Depletion Events, which would explain how the Hg reaches remote polar areas and becomes available for the microbes. Secondly, the presentation of potential benefits of combined chemical and microbiological (in particular -omics approach) studies could also be improved. There is one example given in lines 201-222, however, scientific literature is full of cases (from a variety environments) where the observed environmental phenomena were explained with successful combination of both approaches (e.g. papers by Farooq Azam or Gerhard Herndl). I would also suggest to mention the single-cell (bio)chemical analysis, which can to some extent quantify transformational rates at the species level and more importantly link their metabolic responses to the specific environmental changes. Thirdly, an important link between sympagic communites and global atmospheric change is iron (and ther trace metals) in the sea ice and its role in the fertilization of Polar Oceans during ice melt (plus terrestrial and atmospheric inputs). This could also be explored in the present paper to strenghten the case for the need of microbiochemical studies.
Finally, I am very happy to read paper that encourages scientific efforts for the benefit of both humans and ecosystems and wholeheartedly recommend its publication.
Author Response
I thank the reviewer for her/his positive comments and constructive suggestion. I did my best to incorporate the suggested changes. However, this is an opinion paper which needs to be concise and short. It is not a review and is therefore intentionally incomplete, as it just consists of selected examples. Many additions could be suggested, every reviewer being tempted to make suggestions drawn from her/his own field.
In any case, I agree that a couple of sentences to explain Atmospheric Mercury Depletion Events (AMDEs) which form a large part of the motivation to study mercury chemistry and microbial metabolism in snow, is in order. I therefore added lines 68-72 to detail this aspect.
Regarding -omics approaches, this has indeed been successfully applied in other fields such as marine biochemistry and I guess the reviewer is from that field. I added a paragraph, lines 233-242, to mention that aspect, with some references from marine biochemistry and microbiology as recommended by the reviewer. To better link this aspect with snow microbiology and chemistry, I added the example of organics in snow, a field where some work has been done by snow chemists. To introduce this aspect earlier on, I also added a paragraph lines 174-178, as well as a few words in the introduction (line 36) to mention carbon and organics from the start.
Regarding single-cell biochemical studies, this is indeed quite interesting, but given the state of snow studies, it seems a bit far-fetched and presumptuous on my part to make such a bold suggestion, I therefore abstained, but I thank the reviewer for the interesting suggestion. Lastly, the role of iron is important for the marine food web, but regarding snow itself it probably is not as critical. I therefore think this aspect does not fall in the core of the objectives of this paper, and I believe discussing organics, nitrogen and halocarbons in some detail is sufficient to get my point across.
Reviewer 2 Report
I completely agree on the urge for collaborative studies between microbiologists and chemists to further investigate the functioning of snow ecosystem and thus in order to characterize the impact of climate change at local and large-scale.
This article is indeed highly appropriate for highlighting the lack and the need of collaboration in this field and is therefore particularly relevant in this special issue about snow and ice microbiology.
However, as one of them, I don’t think that microbiologists are willingly ignoring the chemical aspect of the environment and really would like to include microbiological data in bigger environmental picture. But we are still missing so much information to characterize life in such unique habitat. For example you mentioned the potential growth of snow microbes but the turnover of microbes within snow (growth rate and persistence of dead cells) is actually a big bottleneck of snow microbial ecology, which to some extent limits further extrapolation on microbial activities. I would have liked that the article provided more details on specific research axes and/or type of data from both sides that should be targeted in future studies. But I understand that the main objective is to trigger discussion between both communities, and this is highly relevant.
Author Response
Again, I thank the reviewer for her/his constructive comments. These new suggestions have been for the most part addressed in the modifications described in my response to the previous review. In lines 233-242, I propose that coupling “multi-omics methods to state of the art chemical analysis such as FTICR-MS may reveal how microbes modify the chemical composition of the snow and perhaps how this impacts the composition of the atmosphere and climate”. To also address the remark that “I don’t think that microbiologists are willingly ignoring the chemical aspect of the environment” I added the references of Antony et al.. the first author is a microbiologist who has also done some chemistry work, indicating that some significant efforts are starting to be visible in this community.